# Elevated Ozone Reduces the Quality of Tea Leaves but May Improve the Resistance of Tea Plants

**DOI:** 10.3390/plants13081108

**Published:** 2024-04-16

**Authors:** Nuo Wang, Yuxi Wang, Xinyang Zhang, Yiqi Wu, Lan Zhang, Guanhua Liu, Jianyu Fu, Xin Li, Dan Mu, Zhengzhen Li

**Affiliations:** 1Anhui Provincial Key Laboratory of the Biodiversity Study and Ecology Conservation in Southwest Anhui, School of Life Sciences, Anqing Normal University, Anqing 246133, China; 2Key Laboratory of Tea Quality and Safety Control, Ministry of Agriculture and Rural Affairs/Tea Research Institute, Chinese Academy of Agricultural Sciences, Hangzhou 310008, China; 3College of Landscape Architecture, Zhejiang A&F University, Hangzhou 311300, China; 4Tea Research Institute, College of Agriculture and Biotechnology, Zhejiang University, Hangzhou 310058, China

**Keywords:** ozone pollution, tea quality, catechins, BVOCs, chemical defenses

## Abstract

Tropospheric ozone (O_3_) pollution can affect plant nutritional quality and secondary metabolites by altering plant biochemistry and physiology, which may lead to unpredictable effects on crop quality and resistance to pests and diseases. Here, we investigated the effects of O_3_ (ambient air, Am; ambient air +80 ppb of O_3_, EO_3_) on the quality compounds and chemical defenses of a widely cultivated tea variety in China (*Camellia sinensis cv.* ‘Baiye 1 Hao’) using open-top chamber (OTC). We found that elevated O_3_ increased the ratio of total polyphenols to free amino acids while decreasing the value of the catechin quality index, indicating a reduction in leaf quality for green tea. Specifically, elevated O_3_ reduced concentrations of amino acids and caffeine but shows no impact on the concentrations of total polyphenols in tea leaves. Within individual catechins, elevated O_3_ increased the concentrations of ester catechins but not non-ester catechins, resulting in a slight increase in total catechins. Moreover, elevated O_3_ increased the emission of biogenic volatile organic compounds involved in plant defense against herbivores and parasites, including green leaf volatiles, aromatics, and terpenes. Additionally, concentrations of main chemical defenses, represented as condensed tannins and lignin, in tea leaves also increased in response to elevated O_3_. In conclusion, our results suggest that elevated ground-level O_3_ may reduce the quality of tea leaves but could potentially enhance the resistance of tea plants to biotic stresses.

## 1. Introduction

Tropospheric ozone (O_3_), a secondary photochemical pollutant with an oxidant nature, is considered the most widespread air pollutant threatening agricultural production [1,2]. Due to increasing emissions of O_3_ precursors in industrialization, China has become a hotspot of O_3_ pollution, with an increasing rate of 1–3 ppb year^−1^ [3,4,5]. Ozone pollution over China has resulted in an estimated 8.6–32.8% reduction in three major staple crops (rice, maize, wheat), equivalent to a loss of 63 billion U.S. dollars per year [6]. While the deleterious impacts of elevated O_3_ on crop yield have been well documented [6,7,8], our understanding of O_3_ impacts on crop quality lags far behind. In addition, current studies about crop quality focused exclusively on the grain quality of rice, maize, wheat, and soybeans [1]. The influence of elevated O_3_ on other crops, especially for lucrative cash crops, is poorly understood.

Tea (*Camellia sinensis* L.) is a well-known cash crop native to China that has been widely planted in tropical and subtropical areas [9]. Due to its significant economic benefits, tea has been recognized as one of the most important cash crops that plays a significant role in rural development and poverty reduction around the world. China is the world’s biggest tea producer, with 3.39 Mha of tea plantation and around 3 Mt of dry semifinished tea production per year [10], accounting for about 70% of the global tea planting area and more than 40% of global tea production [9]. Primary and secondary metabolites, such as amino acids, catechins, and biogenic volatile organic compounds (BVOCs), account for the flavor and defense capacity of tea leaves and thus determine the quality and economic value of tea production [11]. These metabolites are susceptible to environmental disturbances (such as elevated CO_2_, strong light, and drought), causing fluctuations in chemical concentrations of up to 50% in both positive and negative directions [12]. Despite most of the tea plantations in China being located in areas where O_3_ concentrations exceed the threshold for plant injury (i.e., 40 ppb) [13], very little is known about how elevated O_3_ affects the major chemical compounds in tea [14].

Elevated tropospheric O_3_ may alter phytochemical profiles via several mechanisms. On one hand, reactive oxygen species (ROS) generated by O_3_ can disrupt photosynthesis and inhibit the production of primary and secondary metabolites [15]. On the other hand, O_3_ and ROS can trigger defensive responses and increase the production of antioxidant compounds [16,17]. In general, elevated O_3_ leads to reduced amino acid concentrations due to increased consumption for antioxidant synthesis and repair processes upon O_3_ damage [18]. The responses of phenolics to elevated O_3_ are complex: lignin typically accumulates in plants grown under elevated O_3_ to provide a physical barrier to O_3_ [19], while concentrations of other phenolics (e.g., catechins, condensed tannins) generally increase at a relatively low dose but may decline under a higher dose of O_3_ exposure [20,21]. The effects of O_3_ on BVOCs vary among types of volatiles, with total BVOCs increased but isoprene generally reduced in response to elevated O_3_ [22,23,24]. Ozone impacts on alkaloids are poorly understood; however, the key precursors to alkaloids (polyamines) are reported to be associated with O_3_ tolerance in plants [17].

This study investigated the effects of elevated O_3_ on the phytochemicals that contribute to tea quality and plant resistance in tea leaves. The main quality compounds, including total polyphenols, amino acids, caffeine, and catechins, as well as chemicals with defense functions such as BVOCs, condensed tannins, and lignin, were determined. Leaf quality for green tea was estimated using tea quality indexes. We hypothesized that (1) elevated O_3_ would decrease the concentration of amino acids in tea leaves; (2) elevated O_3_ would enhance the synthesis of secondary metabolites in tea leaves; and (3) impacts of O_3_ on chemical compounds might influence tea quality. 

## 2. Results

### 2.1. Effect of Elevated O_3_ on Main Quality Compounds

Elevated O_3_ exhibited varied influences on the concentrations of key compounds (total polyphenols, amino acids, caffeine, and catechins) that contribute to tea quality. Specifically, elevated O_3_ led to a decrease in amino acids and caffeine concentrations in tea leaves by 39% and 17%, respectively, while showing no significant impact on the concentrations of total polyphenols and total catechins (Figure 1). Impacts of elevated O_3_ on catechins varied depending on their structure. In general, O_3_ decreased the total concentration of ester catechins (ECG, epicatechin gallate; EGCG, epigallocatechin gallate) by 15% while having minimal effects on the total concentration of non-ester catechins (C, catechin; EC, epicatechin; EGC, epigallocatechin) in tea leaves (Figure 1e). Among the detected individual catechin components, we observed significant decreases in concentrations for C (by 22%), EC (by 31%), and ECG (by 15%), with a non-significant decreasing trend for EGCG concentrations in response to elevated O_3_ in tea leaves. Similar to the response observed in total polyphenols, both trolox equivalent antioxidant capacity (TEAC) and ferric reducing antioxidant power (FRAP) also exhibited no significant changes in response to elevated O_3_ (Figure 2). 

Additionally, we calculated commonly used tea quality indices based on the results of the main quality compounds in tea leaves. Our results showed that high concentrations of O_3_ decreased the ratio of total polyphenols to amino acids (TP/AA) and catechin quality index (CQI) (Table 1), suggesting decreased leaf quality for green tea.

### 2.2. Effect of Elevated O_3_ on Chemicals with Defensive Functions

Elevated O_3_ increased the emission of total BVOCs in tea leaves (Figure 3). Our study identified 18 volatiles, classified into five categories: green leaf volatiles (GLVs), aromatics, terpenes, alkanes, and other volatiles (Table 2). Among the detected monomeric BVOCs, O_3_ increased the relative abundances of toluene, (E)-2-hexenal, 6-epi-shyobunol, neophytadiene, and tridecane, while decreasing the relative abundance of 2-ethyl-1-hexanol. Overall, the content of total BVOCs was enhanced by the elevated O_3_, with the relative abundances of GLVs, aromatics, and terpenes increased by 166%, 167%, and 83%, respectively, when compared with controlled plants.

In our study, GLVs, aromatics, and terpenes collectively accounted for over 80% of total BVOCs in tea plants, and their proportions increased significantly in response to elevated O_3_. When compared with controlled tea plants, the proportion of terpenes increased from 39% to 46%, GLVs increased from 7% to 11%, and aromatics increased from 15% to 25% in tea plants exposed to elevated O_3_. Conversely, decreased proportions were observed for alkanes (from 8% to 5%) and others (from 31% to 13%) in tea plants grown under elevated O_3_ conditions, primarily due to the O_3_-induced increase of relative abundances of GLVs, aromatics, and terpenes. 

In addition, elevated O_3_ led to a 35% increase in the concentrations of condensed tannins and a 21% increase in lignin (Figure 4).

## 3. Discussion

### 3.1. O_3_ Impacts on Main Quality Compounds

Amino acids, caffeine, and polyphenols are the three major taste components in tea, contributing to umami, bitterness, and astringency, respectively [11]. Consistent with our first hypothesis, elevated O_3_ decreased the concentration of amino acids in tea leaves. This finding aligns with previous studies on poplars, which also observed decreased amino acids concentrations following O_3_ exposure. This response is likely attributed to the increased demand of amino acids in the detoxification and repair processes following O_3_ injury [18,25]. 

Contrary to our second hypothesis, elevated O_3_ shows varied impacts on secondary compounds. Ozone fumigation resulted in reduced caffeine concentrations in tea leaves, partially supporting previous studies in tea plants that observed decreased caffeine synthesis under stressful conditions [26]. Interestingly, elevated O_3_ showed minimal effects on the concentrations of total polyphenols, aligning with the findings of Da Rosa Santos et al. [16] in *Tibouchina pulchra* (Cham.), where no significant response of total polyphenols to increased O_3_ concentration was observed. Nevertheless, our results diverge from other studies reporting either increased or decreased foliar total polyphenols in response to elevated O_3_ [27,28]. The inconsistent response of polyphenols across different studies might be attributed to the concept of O_3_-induced hormesis in plants. This theory suggests that the production of antioxidative compounds may increase at low-to-moderate O_3_ exposures but decline when the defense response is overwhelmed under higher O_3_ exposure [21,29]. In the present study, antioxidant capacity in tea leaves mirrored the response of phenolics and showed no response to elevated O_3_, which potentially supports the concept of hormesis. Additionally, other factors, including plant species, genotypes, plant development status, and the duration of O_3_ fumigation, may also contribute to the variable responses of total polyphenols to elevated O_3_ treatments [20]. 

It is worth noting that, while the total polyphenols and overall catechins concentrations exhibited a minimal response to elevated O_3_, concentrations of C, EC, and ECG reduced significantly under elevated O_3_ treatment. According to the results of the TEAC assay, the scavenging capacity of the five main catechin compounds in tea follows this order: EGCG > EGC > ECG > EC > C [30]. In this context, O_3_-induced decline in the concentrations of C, EC, and ECG may suggest that tea plants tend to prioritize the synthesis of compounds with relatively higher radical scavenging capacity, such as EGC and EGCG, under O_3_ stresses. Moreover, the concentrations of ester catechins—serving as the primary astringent compounds in tea—decrease with increasing O_3_ concentration, which may lead to changes of tea quality. 

Quality indices, specifically represented by TP/AA and CQI, provide a quick assessment of the taste and overall quality of tea based on changes of flavoring compounds. Our results showed that both the TP/AA ratio and CQI value decreased in response of elevated O_3_. The TP/AA ratio is widely acknowledged as a reliable marker of tea quality, influencing the taste by making tea either brisk and mellow with a lower TP/AA ratio or more astringent with a higher ratio [31]. Meanwhile, the value of CQI has a relatively limited scope of application compared to the TP/AA ratio, primarily due to the varied catechin accumulation characteristics among species during shoot elongation [32]. In our present study, however, the use of tea cutting seedlings from the same cultivar ensured no variation in catechin accumulation characteristics among treatment plants. Consequently, the CQI value becomes a valuable tool to discern the impact of O_3_ fumigation on tea quality, with an increased CQI value indicating decreased tea quality upon O_3_ stresses. 

Collectively, our results suggest that elevated O_3_ may decrease the quality of tea leaves, as predicted in our third hypothesis. Specifically, the taste profile of green tea may be altered by elevated O_3_, potentially leading to a decrease in umami, bitterness, and astringency. Given the potential interactions among taste compounds in modifying taste intensity [33], future studies incorporating sensory analysis can be conducted to clearly identify the impacts of elevated O_3_ on the taste of tea.

### 3.2. O_3_ Impacts on Chemicals with Defensive Functions

Biogenic volatile organic compounds generally function as communication media between plants and insects [34]. In the present study, O_3_ was found to increase the release of BVOCs categories that have herbivore defense and/or predator attractive functions, represented as elevated loads of GLVs, terpenes, and aromatic compounds. Our results are in agreement with studies on broccoli and hybrid aspens, which reported higher amounts of GLV and terpenes when expose to elevated O_3_ [35,36]. However, our findings contrast with one study on *Brassica napus* that observed a decreased emission of terpenes under elevated O_3_ conditions [37]. It has been well documented that GLVs serve as non-specific signals of on-going damage that can be utilized by natural predators [38,39], while terpenes function as highly specific defense signals that synthesized through specialized biosynthetic processes [40]. Therefore, the O_3_-induced increase in emissions of GLVs and terpenes may suggest an enhanced defense capacity of tea plants against insect herbivores.

It is noteworthy that elevated O_3_ significantly increased the relative abundance of (*E*)-2-hexenal, a GLV component acting as an infochemical for insects, in tea leaves. (*E*)-2-hexenal has been reported to attract both insect pests and natural predators within tea plantations [41,42]. Using both Y-tube olfactometer and field trapping tests, Mu et al. [43] demonstrated the role of (*E*)-2-hexenal in attracting tea shoot volatiles to tea green leafhoppers (*Empoasca vitis Gothe*). In the electroantennogram responses and the wind tunnel bioassays, Han and Chen [44] found that several natural predators, including *Chrysopa sinica* and *Coccinella septempunctata*, exploit (*E*)-2-hexenal as a long-range cue to locate and approach tea aphids (*Aphidius* sp.). Therefore, the O_3_-induced enhanced emission of (*E*)-2-hexenal in our study indicates that elevated O_3_ may exert an unpredictable influence on the relationship between tea plants and insect pests.

In line with findings in other plant species [19,20,45], our study observed increased concentrations of condensed tannins and lignin in tea plants exposed to elevated O_3_ compared to those grown in ambient air. These results supported the notions that plants defend against O_3_ stresses by accumulating antioxidative components and reinforcing structural barrier to O_3_ [19]. However, some studies have reported unchanged or even reduced condensed tannins in response to elevated O_3_ [36,45,46]. This variation has been attributed to the inhibition of carbon-based defensive compounds synthesis due to heavily suppressed plant photosynthetic carbon fixation under elevated O_3_ [47]. As two, well-studied defense chemicals in plants, condensed tannins and lignin are believed to reduce herbivory by increasing toxicity or decreasing palatability of host plants to insect herbivores [14,45,46]. They are also responsible for plant protection against pathogens [48]. An ozone-induced increase in condensed tannins and lignin concentrations has been reported to impact the performance of herbivores [45,49].

## 4. Materials and Methods

### 4.1. Experimental Site and Plant Materials

The study was conducted within open-top chambers (OTCs; 3 m in height, 4 m in diameter, octagonal shape, glass-covered) at the Tea Research Institute, Chinese Academy of Agricultural Sciences located in Hangzhou, China (30°10′ N, 120°5′ E, 19 m a.s.l.), from early spring to early summer of 2023. The experimental site features a subtropical monsoon climate with a mean annual air temperature of approximately 17.0 °C and a mean annual precipitation of exceeding 1500 mm. The widely cultivated tea variety ‘BaiYe 1 Hao’ was used for analysis. Tea cutting seedlings were planted in a sterilized germination mix (vermiculite: perlite: nutritive soil = 1:1:3) and allowed to grow for two years before being transplanted into 5-gallon pots in October 2022. On February 28th, tea plants were transferred to OTCs for a 7-day adaptation period. Subsequently, 16 tea plants with similar growth conditions were selected and randomly distributed into four OTCs for different O_3_ treatments, with a total of four plants per OTC. Throughout the entire experimental period, tea plants were supplied with water as needed and maintained through standardized horticultural practices. At the time of the experiment, the tea plants were three years old.

### 4.2. Experimental Design

The experiment set up two O_3_ concentration levels (Am, ambient air; EO_3_, ambient air + 80 ppb O_3_) with two replicated OTCs per O_3_ treatment. Ozone was generated from pure oxygen using an O_3_ generator (CFG-20, Sankang Co., Jinan, China), and a mixture of O_3_ and ambient air was ventilated into each OTC through Teflon tubes. Continuous monitoring of O_3_ concentration in the OTCs was achieved using a UV photometric O_3_ detector (Model 49i, Thermo Fisher Scientific, Franklin, MA, USA). The flow rate of O_3_ was controlled by an automatic controller (SC200, SevenStar Flow Co., Kunshan, China) to achieve the target concentration. Ozone fumigation was implemented for 8 h per day (from 9:00 to 17:00) over a duration of 96 days (from 6 March to 10 June 2023), with temporary suspension on days of precipitation. Throughout the fumigation period, the mean daily O_3_ concentrations within the OTCs under Am and EO_3_ treatment were 22.72 ± 1.28 ppb and 91.92 ± 1.43 ppb, respectively. Correspondingly, the AOT40 values (i.e., the accumulated O_3_ exposure over a threshold of 40 ppb in daily hours) for the Am and EO_3_ treatments during the entire experimental period were 1.57 ± 0.58 ppm h and 56.36 ± 1.60 ppm h, respectively.

### 4.3. Sample Collection

Following O_3_ fumigation, two plants were randomly chosen from each OTC for sampling. The shoot tips (one bud and two leaves) were collected from tea plants, promptly immersed into liquid nitrogen, and transported to the laboratory within 2 h. Each sample was divided into two halves for distinct analyses. The first half was vacuum dried, pulverized by ball milling, and then stored at −20 °C for analyses of non-volatile chemical compounds. The other half was stored at −80 °C to facilitate the analyses of biogenic BVOCs.

### 4.4. Chemical Analyses

Concentrations of total polyphenols were analyzed using a Folin–Ciocalteu assay with a slight modification [21]. In brief, 20–30 mg of leaf powder was extracted with 1.5 mL methanol, with 30 min of sonication. Properly diluted supernatant was then mixed with 10-fold diluted Folin–Ciocalteu reagent and 10% sodium carbonate water solution. The mixture was incubated in the dark for 30 min, and the absorbance at 760 nm was measured using a spectrophotometer (SpectraMax M2, Molecular Devices, Sunnyvale, CA, USA). The concentration of each sample was computed relative to standardization with gallic acid.

Free amino acids were quantified using the ninhydrin method modified from the Standardization Administration of China, GB/T 8314-2013 [50]. Leaf powder (20–30 mg) was extracted with boiled distilled water (1 mL) for 5 min. A subsample of the supernatant (100 μL) was mixed with 50 μL of 2% ninhydrin solution (augmented with 0.08% stannous oxide) and 50 μL of phosphate buffer (pH 8.0). After being heated in a boiling water bath for 5 min to facilitate the reaction, absorbance of the mixture was determined at 570 nm, and amino acid concentrations were calculated against theanine authentic standards.

Caffeine and the major individual catechins (C, EC, EGC, ECG, EGCG) were extracted into methanol and quantified using a high-performance liquid chromatography (HPLC; Alliance e2695, Waters, Milford, MA, USA) with a photodiode array detector (ACQUITY 2998, Waters, Milford, MA, USA) as described by Li et al. [30]. The retention time and chromatogram of each analyte are detailed in Appendix A and Appendix A. Concentrations of caffeine and individual catechins were quantified based on their peak areas with calibration against authentic standards. Total catechins were determined by summing the concentrations of the five individual catechins. Ester catechins were determined by summing the concentrations of ECG and EGCG. Non-ester catechins were determined by summing the concentrations of C, EC, and EGC.

Condensed tannins were determined using the acid–butanol method with modifications [51]. In brief, condensed tannins were extracted from 20–30 mg of leaf powder into 1.5 mL 70% acetone solution (include 10 mM ascorbic acid) with 30 min of sonication. A subsample of the supernatant (100 μL) was mixed with 400 μL of 70% acetone solution, 2 mL N-butanol/hydrochloric acid (95:5) solution, and 100 μL acidified 2% ferric ammonium sulfate solution (dissolved with 2N hydrochloric acid), followed by a 50 min boiling water bath. Condensed tannins were quantified by measuring the absorbance at 550 nm with standardization against proanthocyanins. Lignin concentrations were determined by the thioglycolic acid method following the steps described in Suzuki et al. [52], with alkali lignin as standards.

Quantification of BVOCs was performed using a gas chromatograph-mass spectrometer (GC-MS; 8890/5977B GC-MS system, Agilent Technologies, Santa Clara, CA, USA). The organic solvent was extracted with ethyl acetate (1:30,000) containing n-octanol as internal standard. Samples (2 μL) were injected into an HP-5MS chromatographic column (30 m × 0.25 mm, 1909IS-433UI, Agilent Technologies, Santa Clara, CA, USA). The injection temperature was 250 °C, and a temperature gradient of 5 °C/min from 40 °C (hold 3 min) to 250 °C was employed. The mass spectrometer (MS/MS) was operated in the electronic ionization (EI) mode with 70 eV of electron energy. All MS data were collected from 40 to 400 *m*/*z* [53]. The relative content of BVOCs was calculated by dividing the total ion current peak areas by the peak area of the internal standard. The sample chromatograms are shown in Appendix A.

### 4.5. Antioxidant Activity

Trolox equivalent antioxidant capacity (TEAC) of tea samples was determined using DPPH radical and ABTS radical cation assays following the steps described in Li et al. [30]. The ferric reducing antioxidant power (FRAP) was measured by incubating a mixture of 100 μL leaf extract and 900 μL FRAP reagent (contained 8.3 mM 2,4,6-tripyridyl-s-triazine and 16.7 mM FeCl_3_) at 37 °C for 10 min, followed by measuring the absorbance at 593 nm. 

### 4.6. Assessment of Tea Quality Indexes

The ratio of total polyphenols to amino acids (TP/AA) reflects the balance between mellow tastes and astringency. Typically, the TP/AA ratio is negatively correlated with the quality of green tea, where a lower TP/AA ratio leads to a more brisk and mellow, but less bitter in taste [54]. Another quality index, catechin quality index (CQI), is calculated as the ratio of (EGCG + ECG) to EGC and is negatively correlated with the tenderness and overall quality of green tea [55]. 

### 4.7. Statistical Analyses

SPSS 26.0 (SPSS Inc., Apache Software Foundation, Chicago, IL, USA) was used for statistical analysis of the calculated data under different treatments. To assess the effects of elevated O_3_ on the concentrations of main chemical compounds, tea quality indexes, relative abundances of BVOCs, and defense chemical compounds, the data were subjected to a one-way analysis of variance (one-way ANOVA). The Shapiro–Wilk normality test was applied to verify normality and homogeneity of variance. Differences were considered significant if *p* < 0.05. All quantitative data were expressed as mean ± *SD* (*n* = 4 trees).

## 5. Conclusions

Our study revealed that elevated O_3_ altered the concentrations of both quality and defense compounds in tea leaves. Impacts of elevated O_3_ on the quality of tea leaves is primarily attributed to decreased amino acid and caffeine concentrations and an altered catechins composition. Ozone was found to have decreased TP/AA and CQI, signifying a decrease in the quality of green tea. Furthermore, elevated O_3_ increased the release of BVOCs compounds with herbivore defense and/or predator-attractive functions in tea, as evidenced by the increased relative abundance of GLVs, aromatics, and terpenes. Elevated O_3_ also enhanced the accumulation of condensed tannins and lignin, indicating an improved resistance of tea plants to insect pests. In summary, our findings suggest that ozone concentrations of around 90 ppb reduce the quality of tea but may enhance the ability of tea plants to resist pests and diseases. Future studies should prioritize investigating the dose–response relationships between elevated tropospheric O_3_ and tea quality to accurately estimate the impacts of O_3_ pollution in tea production. Additionally, developing methods to mitigate the adverse effects of O_3_ pollution requires a deeper understanding of the mechanisms underlying O_3_ impacts on tea quality compounds. 

## Figures and Tables

**Figure 1 plants-13-01108-f001:**
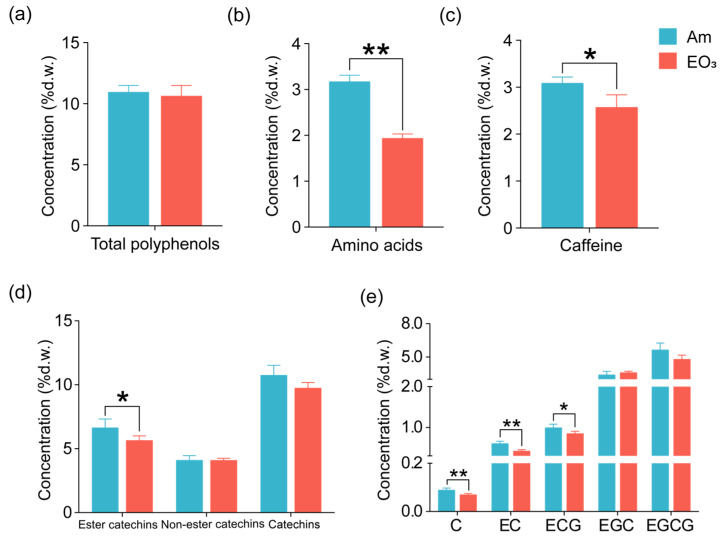
Effect of elevated O_3_ on main quality compounds in tea leaves. (**a**) Total polyphenols; (**b**) amino acids; (**c**) caffeine; (**d**) ester catechins (Σ ECG, EGCG), non-ester catechins (Σ C, EC, EGC) and total catechins; (**e**) individual catechin compounds. Each bar represents the mean value ± *SD* (n = 4 trees). % d.w. = percentage of foliar dry weight. Stars indicate significant differences between O_3_ treatments (*, 0.01 < *p* < 0.05; **, *p* < 0.01). Am, ambient air; EO_3_, ambient air +80 ppb of O_3_; C, catechin; EC, epicatechin; ECG, epicatechin gallate; EGC, epigallocatechin; EGCG, epigallocatechin gallate.

**Figure 2 plants-13-01108-f002:**
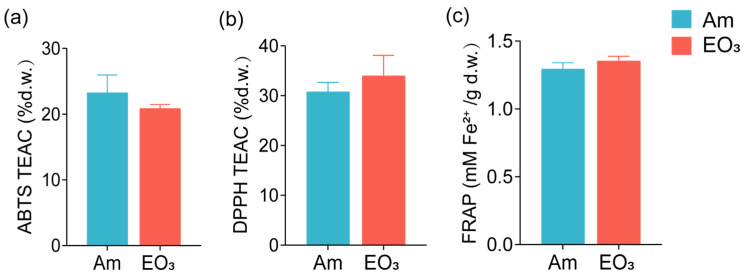
Effect of elevated O_3_ on the antioxidant capacity of tea leaves. (**a**) Trolox equivalent antioxidant capacity (TEAC) in 2,20-azinobis (3-ethylbenzothiazoline-6-sulfonic acid) (ABTS) radical scavenging assay; (**b**) TEAC in 2,2-diphenyl-1-picrylhydrazyl (DPPH) radical scavenging assay; (**c**) ferric reducing antioxidant power (FRAP). Each bar represents the mean value ± *SD* (*n* = 4 trees). % d.w. = percentage of foliar dry weight. Am, ambient air; EO_3_, ambient air +80 ppb of O_3_.

**Figure 3 plants-13-01108-f003:**
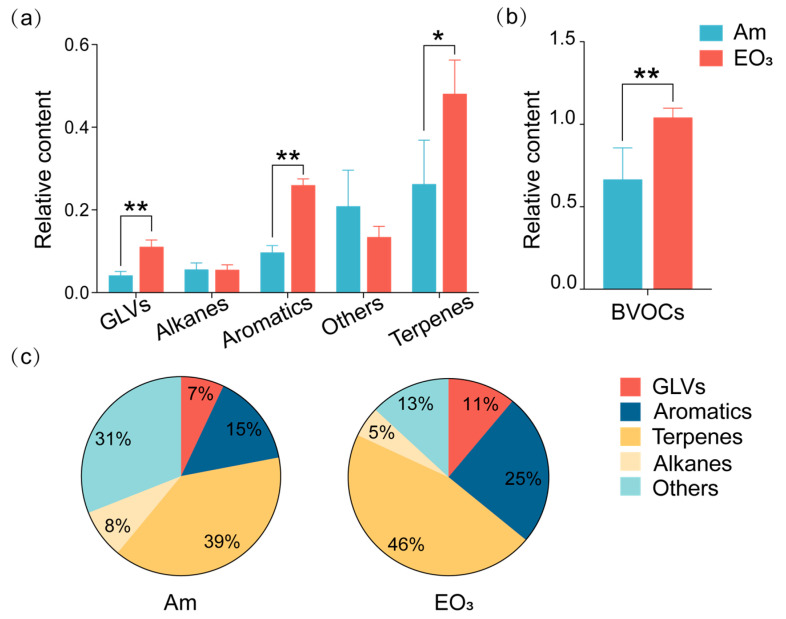
Effect of elevated O_3_ on major BVOC classes in tea leaves. (**a**) Relative content of five major BVOC categories; (**b**) relative content of total BVOCs; (**c**) proportional representation of five major BVOC categories in total measured BVOC pool in tea leaves. Each bar represents the mean value ± *SD* (*n* = 4 trees). Stars above the bars indicate significant differences between O_3_ treatments (*, 0.01 < *p* < 0.05; **, *p* < 0.01). Am, ambient air; EO_3_, ambient air +80 ppb of O_3_; GLVs, green leaf volatiles.

**Figure 4 plants-13-01108-f004:**
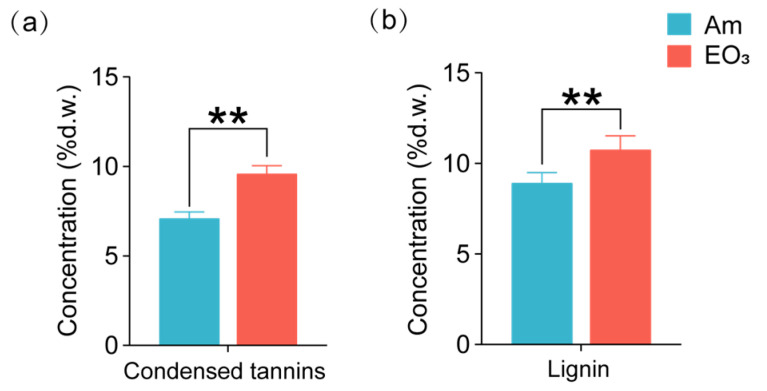
Effect of elevated O_3_ on major defensive metabolites in tea leaves. (**a**) Condensed tannins; (**b**) lignin. Each bar represents the mean value ± *SD* (*n* = 4 trees). % d.w. = percentage of foliar dry weight. Stars indicate significant differences between O_3_ treatments (**, *p* < 0.01). Am, ambient air; EO_3_, ambient air +80 ppb of O_3_.

**Table 1 plants-13-01108-t001:** Effect of elevated O_3_ on tea quality indexes.

Tea Quality Index	Am	EO_3_
TP/AA	3.45 ± 0.10 ^b^	5.50 ± 0.45 ^a^
CQI	1.96 ± 0.23 ^a^	1.57 ± 0.07 ^b^

Values are depicted as the mean ± *SD* (*n* = 4 trees). Different lower-case letters denote significant differences between O_3_ treatments (*p* < 0.05). Am, ambient air; EO_3_, ambient air +80 ppb of O_3_; TP/AA, the ratio of total polyphenols to amino acids; CQI, catechin quality index.

**Table 2 plants-13-01108-t002:** Effect of elevated O_3_ on major volatile organic compounds in tea leaves.

Volatile Class	Individual Compounds	Retention Time (min)	Am	EO_3_
Green leaf volatiles	E-2-hexenal	6.632	0.025 ± 0.010 ^b^	0.106 ± 0.014 ^a^
	2-Ethyl-1-hexanol	11.901	0.016 ± 0.008 ^a^	0.005 ± 0.001 ^b^
Aromatics	Toluene	5.303	0.050 ± 0.007 ^b^	0.228 ± 0.008 ^a^
	Benzyl alcohol	12.156	0.005 ± 0.003 ^a^	0.003 ± 0.001 ^a^
	2,4-Di-tert-butylphenol	25.103	0.042 ± 0.006 ^a^	0.029 ± 0.009 ^a^
Terpenes	Trans-Furanic linalool oxid	13.591	0.005 ± 0.003 ^a^	0.005 ± 0.001 ^a^
	Geraniol	18.901	0.003 ± 0.001 ^a^	0.003 ± 0.001 ^a^
	Neophytadiene	32.261	0.172 ± 0.064 ^b^	0.362 ± 0.059 ^a^
	Phytol	37.536	0.078 ± 0.024 ^a^	0.091 ± 0.020 ^a^
	6-epi-shyobunol	28.565	0.005 ± 0.002 ^b^	0.021 ± 0.004 ^a^
Alkanes	Undecane	11.312	0.002 ± 0.001 ^a^	0.003 ± 0.001 ^a^
	Dodecane	12.635	0.010 ± 0.005 ^b^	0.015 ± 0.003 ^a^
	Tridecane	17.246	0.009 ± 0.001 ^a^	0.007 ± 0.001 ^a^
	Pentadecane	19.088	0.032 ± 0.009 ^a^	0.027 ± 0.007 ^a^
	1-Methoxyadamantane	16.628	0.003 ± 0.001 ^a^	0.004 ± 0.001 ^a^
Others	2,4-Dimethyl-1-heptanol	13.976	0.007 ± 0.002 ^a^	0.006 ± 0.001 ^a^
	Hexyl octyl ether	14.151	0.002 ± 0.001 ^a^	0.001 ± 0.000 ^a^
	Cyclohexanol	7.442	0.201 ± 0.076 ^a^	0.128 ± 0.021 ^a^

Values are depicted as the mean ± *SD* (*n* = 4 trees). Different lower-case letters denote significant differences between O_3_ treatments (*p* < 0.05). Am, ambient air; EO_3_, ambient air +80 ppb of O_3_.

## Data Availability

Original data are available upon request from the corresponding author.

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
