# Peer review of "Elevated Ozone Reduces the Quality of Tea Leaves but May Improve the Resistance of Tea Plants"

_plants, 2024, doi:10.3390/plants13081108_

Round 1

Reviewer 1 Report

Comments and Suggestions for Authors

The manuscript submitted by Wang et al. describes the impacts of elevated ambient O3 levels on tea plants with respect to tea leaves quality and resistance of tea plants. The topic is interesting and I think it is a nice piece of work, attractive for readers of Plants journal. The work is described clearly. However, I have few comments which might enhance the original draft.

1. I suggest to move the chapter Methods before results. I  do not understand why the authors opted for unusual arrangement (unless it is a specific requirements by the journal editors) placing the Methods after the Results and Discussion.

2. The acronyms should be explained the very first time they appear in the text, which is not the case now. It should be revised thoroughly.

3. The TP/AA and CQI must be explained.

4. Fig. 1 - the desription of y-axis is not clear, it should be explained

5. Fig. 2 - What is the meaning of "Cumulative relative contents" describing y-axis? Furthermore, the units should be added.

6. Table 2 - the values should be rounded appropriately. Furthermore - what is the exact meaning of "retention"?

7. Chapter 2.2. - Why do the authors write the individual BVOCs with capital letters? It should be corrected.

8. Introduction - line 43 - "increasing speed of 1–3 ppb O3 per year" should be replaced by some more suitable wording. ppbs are not correct units for speed, they are correct for concentrations. Hence "increase in concentrations" sounds more proper in this case.

9. Introduction - line 62 - "50% alterations in concentrations" - it is unclear of what concentrations exactly the authors mean in this context - it should be completed

10. Introduction - The hypotheses - I believe that it would be better to divide the first hypothesis into two as these are actually two different points. Hence it would be appropriate to cast three insted of two hypotheses.

Comments on the Quality of English Language

OK

Author Response

The manuscript submitted by Wang et al. describes the impacts of elevated ambient O3 levels on tea plants with respect to tea leaves quality and resistance of tea plants. The topic is interesting and I think it is a nice piece of work, attractive for readers of Plants journal. The work is described clearly. However, I have few comments which might enhance the original draft.

  1. I suggest to move the chapter Methods before results. I do not understand why the authors opted for unusual arrangement (unless it is a specific requirements by the journal editors) placing the Methods after the Results and Discussion.

Response: No need to revise because it is a specific requirement by the journal.

  1. The acronyms should be explained the very first time they appear in the text, which is not the case now. It should be revised thoroughly.

Response: Revised. See line 96-97, 101-104 and other places.

  1. The TP/AA and CQI must be explained.

Response: Revised. See line 106-107.

  1. Fig. 1 - the description of y-axis is not clear, it should be explained

Response: We changed the description of y-axis to “Chemical concentration (% d.w.)”. In the figure legend we explained that “% d.w. = percentage of foliar dry weight”.

  1. Fig. 2 - What is the meaning of "Cumulative relative contents" describing y-axis? Furthermore, the units should be added.

Response: We unified the phrase to “Relative content” in the manuscript. We also explained the method used to calculated relative content of BVOCs and no unit is needed according to the method (see line 335-336).

  1. Table 2 - the values should be rounded appropriately. Furthermore - what is the exact meaning of "retention"?

Response: Corrected.

  1. Chapter 2.2. - Why do the authors write the individual BVOCs with capital letters? It should be corrected.

Response: Revised. See line 130-131.

  1. Introduction - line 43 - "increasing speed of 1–3 ppb O3 per year" should be replaced by some more suitable wording. ppbs are not correct units for speed, they are correct for concentrations. Hence "increase in concentrations" sounds more proper in this case.

Response: We revised the sentence. See line 43: “with an increasing rate of 1–3 ppb year-1”.

  1. Introduction - line 62 - "50% alterations in concentrations" - it is unclear of what concentrations exactly the authors mean in this context - it should be completed.

Response: We rewrote the sentence. See line 61-62: “causing fluctuations in chemical concentrations of up to 50% in both positive and negative directions”.

  1. Introduction - The hypotheses - I believe that it would be better to divide the first hypothesis into two as these are actually two different points. Hence it would be appropriate to cast three insted of two hypotheses.

Response: We revised the hypotheses as suggested (line 84-86) and relative sentences in the discussion section (line 164-171, 210).

Reviewer 2 Report

Comments and Suggestions for Authors

The article titled "Elevated ozone reduces the quality of tea leaves but may improve the resistance of tea plants" can be published in Plants after corrections. Environmental pollution is a huge problem for crops. The authors checked the effect of ozone on the content of primary and secondary metabolites in tea leaves. The results suggest a significant impact of ozone on the production of metabolites and reduction in tea quality. Stress conditions reduce the production of amino acids and caffeine and have no significant effect on the production of antioxidant metabolites. For a complete picture, antioxidant activity tests are missing. Studying this activity could confirm the authors' hypotheses and would be a good complement to the ongoing research. I ask the authors to complete the research. They should enrich the publication and provide information about the impact of ozone on an important area of plant defense. I propose to supplement the research using at least three tests examining different aspects of activity: DPPH, ROS and chelating activity.

Other comments:

In the Results section of Figure 1, please provide an explanation of the abbreviations Am and EO3.

In the conclusions section, I ask the authors to formulate the goals of further research. In what direction should further research go?

In supplementary materials, please paste sample chromatograms and UV spectra of the tested compounds and reference substances.

Author Response

The article titled "Elevated ozone reduces the quality of tea leaves but may improve the resistance of tea plants" can be published in Plants after corrections.Environmental pollution is a huge problem for crops.The authors checked the effect of ozone on the content of primary and secondary metabolites in tea leaves.The results suggest a significant impact of ozone on the production of metabolites and reduction in tea quality.Stress conditions reduce the production of amino acids and caffeine and have no significant effect on the production of antioxidant metabolites.For a complete picture, antioxidant activity tests are missing.Studying this activity could confirm the authors hypotheses and would be a good complement to the ongoing research.I ask the authors to complete the research.They should enrich the publication and provide information about the impact of ozone on an important area of plant defense.I propose to supplement the research using at least three tests examining different aspects of activity: DPPH, ROS and chelating activity.

Response: Thanks for this valuable suggestion. We estimated the trolox equivalent antioxidant capacity (TEAC) using DPPH and ABTS assays, and ferric reducing antioxidant power (FRAP) in tea leaves. In our study, DPPH TEAC, ABTS TEAC, and FRAP values showed no significant changes in response to elevated O3. We believe that results of these three parameters are enough to illustrate the impacts of elevated O3on the antioxidant capacity of tea leaves. Relative results have been added to the results (Fig. 2, line101-104), discussion (line 181-184, 188-196) and method section (line 338-343).

Other comments:

In the Results section of Figure 1, please provide an explanation of the abbreviations Am and EO3.

Response: We add the explanation of Am and EO3in the capital of Figure 1 and other figures/tables.

In the conclusions section, I ask the authors to formulate the goals of further research. In what direction should further research go?

Response: We added the direction of further research in the conclusions section (see line 370-374).

In supplementary materials, please paste sample chromatograms and UV spectra of the tested compounds and reference substances.

Response: Added. See Fig. S1-S2.

Round 2

Reviewer 1 Report

Comments and Suggestions for Authors

I believe that the previous version has been enhanced and that the manuscript can be now accepted for publication.

Author Response

No need to revise.

Reviewer 2 Report

Comments and Suggestions for Authors The authors have corrected the article in accordance with the comments. The article can be published in its current form.    

Author Response

No need to revise.